# Bosonization in $R$-paraparticle Luttinger models

**Dennis F. Salinel** and **Kristian Hauser A. Villegas**[†]

National Institute of Physics, University of the Philippines Diliman, Quezon City 1101,
Philippines

⋆ dfsalinel@up.edu.ph , † kavillegas1@up.edu.ph

## Abstract

We reintroduce the parafermion-paraboson classification in $R$-paraparticles in terms of their average occupation numbers, analogous to Green's parastatistics. The notion of $p$-order in $R$-parafermions is also redefined as the maximum number of particles that can occupy a quantum state. An example of an order-2 $R$-parafermion with $m = 2$ internal degrees of freedom is presented, which obeys an exclusion principle that is not Pauli's. The interacting $R$-parafermions are studied in the context of bosonization. Specifically, we show that while density waves are generally bosonic in nature and that flavor-charge separation naturally occurs for any one-dimensional $R$-parafermion system described by the Luttinger model, flavor waves do not always satisfy bose statistics. Comparison of the partition functions further show that only ($p = 1$)-ordered $R$-parafermions are compatible with the bosonization procedure in the low-energy limit. Based from these results, we discuss a potential realization of $R$-parafermion signatures in one-dimensional systems.

# 1 Introduction

Conventionally, particles are categorized as either bosons or fermions; their difference lies primarily with their inherent exchange statistics, where the boson (fermion) wavefunction is symmetric (antisymmetric) under particle exchange. An exception to this dichotomy are anyons [1, 2], existing only in two dimensions [3, 4] and where the particle's wavefunction pick up an arbitrary phase under an exchange of particles.

Outside anyons, multiple theories of exotic particles that do not follow bosonic or fermionic statistics have also been proposed: all of which have not yet been observed experimentally. Probably the most well-studied are Green's paraparticles [5], which to this day garners opposing opinions on its actual existence in nature [6–8]. While superselection rules [9] and no-go theorems [10, 11] tell us that under certain assumptions Green's paraparticles can always be re-expressed in terms of ordinary bosons and fermions [12, 13], these studies do not explicitly rule out the possibility of them existing in nature. The field has become purely theoretical, with paraparticles being explored through their thermodynamic signatures [14], using first-quantized parastatistics theories [8, 52–54] and para-manifolds [15], and as potential candidates for dark matter [16–18]. To date, the only experimental efforts to studying paraparticles are trapped ion simulations of para-bose and para-fermi oscillators [19–22].

Recently, an alternative formulation was proposed which makes use of generalized commutation relations due to a four-tensor $R$ [23]. In this "$R$-paraparticle" formulation, quasiparticle excitations with parastatistical behavior can be constructed using nonlocal string operators from an underlying set of local spin operators. This approach evades the no-go theorems [10, 11] which assumes that excitations are only created by local operators. Moreover, the superselection rules [9] are inherently satisfied by quasiparticle excitations. This construction from local spin operators opens the possibility of realizing emergent paraparticles in condensed matter systems and in quantum simulations using trapped ions [24]. Furthermore, emergent $R$-paraparticles realized in higher dimensional condensed matter systems enable a novel way of nonlocal secret communication, as proposed in [6], which demonstrates their fundamental physical distinction from ordinary fermions and bosons, and provides an experimental way to observe their exotic exchange statistics. However, the discussions in [23] primarily focus on free paraparticle systems. In contrast, conventional bose and fermi systems exhibit a wealth of collective excitations and emergent phases arising from particle interactions. This naturally raises the question of how interactions might influence the behavior of paraparticles, making it an intriguing direction for further investigation.

The Luttinger model [25–28], which describes a one-dimensional fermionic gas at low temperatures, plays a significant role in understanding interacting quantum systems. A key feature of this model is bosonization, where fermionic degrees of freedom collectively behave as bosonic modes. This allows the system's Hamiltonian to be rewritten purely in terms of bosonic operators, making the originally interacting fermionic system exactly solvable [29]. As a result, the model exhibits several non-Fermi liquid properties, including spin–charge separation [30, 31], power-law decay of correlation functions [26, 27, 32], and heightened sensitivity to impurities [33–35].

In this paper, we study a generalized version of the Luttinger model where we extend the conventional fermionic operators to $R$-paraparticle operators. This approach of extending the operators to $R$-paraparticle operators has recently been explored in the context of para-SYK

models [36, 37]. We show that bosonization also occurs for $R$-paraparticles with fermi-like average occupation numbers [14], which we call "$R$-parafermions" in analogy to Green's paraparticles [5, 9]. This presents a method in which interacting $R$-paraparticle systems can be probed using standard methods of solving bosonic systems [29]. Before proceeding to the remainder of the study, we first clarify that the term "parafermion" should not be confused with generalizations of Majorana fermions [38–40], which unfortunately share the same terminology in condensed matter literature. For instance, the work in [41] also addresses the bosonization of parafermions; however, the parafermions considered there are those related to Majorana fermions and are not the $R$-parafermions considered in this work.

The paper shall be organized as follows. In Sec. 2, we review the $R$-paraparticle formulation and clarify how $R$-parabosons and $R$-parafermions are classified based on their thermodynamics and exclusion statistics. We show that $R$-parafermions obey Pauli's exclusion principle or one of its generalizations, prompting us to define $p$-orderedness in the $R$-paraparticle formalism. In Sec. 3, $R$-parafermionic density wave excitations are shown to be bosonic, independent of the number of degrees of freedom of the $R$-paraparticle, and are completely decoupled to flavor waves. However, for systems with two internal degrees of freedom, flavor waves, which are generalizations to spin waves, are bosonic only for specific $R$-parafermion species. More importantly, we show that bosonization only applies at low energies for $R$-paraparticles that satisfy Pauli exclusion principle, as $R$-parafermions obeying higher-order exclusion principles generate partition functions that do not match in the $R$-parafermionic and bosonic bases. These results point to possible experimental realizations of $R$-paraparticle signatures in one-dimensional systems by searching for signs of flavor-charge separation [31] in the absence of magnon excitations [42], as will be discussed in Sec. 4. Finally, we summarize the contents of this paper and provide possible extensions of this work in Sec. 5.

## 2  R-paraparticles

### 2.1  Formulation

The $R$-paraparticle formulation introduced in [23] works as follows. Let $R_{cd}^{ab}$ ($1 \le a, b, c, d \le m \in \mathbb{Z}_+$) be a four-tensor that satisfies

$$\sum_{c,d} R_{cd}^{ab} R_{ef}^{cd} = \delta_{ae} \delta_{bf}, \tag{1a}$$

$$\sum_{g,h,i} R_{gh}^{ab} R_{if}^{hc} R_{de}^{gi} = \sum_{g',h',i'} R_{g'h'}^{bc} R_{di'}^{ag'} R_{ef}^{i'h'}, \tag{1b}$$

the second equation being the constant Yang-Baxter equation (YBE) [43–45]. A paraparticle that is described by a unique $R_{cd}^{ab}$ then satisfies the generalized commutation relations (GCR)

$$\hat{\psi}_{i,a} \hat{\psi}_{j,b}^+ = \sum_{cd} R_{bd}^{ac} \, \hat{\psi}_{j,c}^+ \hat{\psi}_{i,d} + \delta_{ab} \delta_{ij}, \tag{2a}$$

$$\hat{\psi}_{i,a}^+ \hat{\psi}_{j,b}^+ = \sum_{cd} R_{ab}^{cd} \, \hat{\psi}_{j,c}^+ \hat{\psi}_{i,d}^+, \tag{2b}$$

$$\hat{\psi}_{i,a} \hat{\psi}_{j,b} = \sum_{cd} R_{dc}^{ba} \, \hat{\psi}_{j,c} \hat{\psi}_{i,d}, \tag{2c}$$

where $\hat{\psi}_{i,a}^+$ ($\hat{\psi}_{i,a}$) creates (annihilates) a particle at mode $i$ with an internal or flavor degrees of freedom indexed by $a$. The integer $m$ denotes the maximum number of internal degrees

of freedom (which need not be related to any observable [13]) considered in the interaction. The creation and annihilation operators satisfy $\hat{\psi}^+_{i,a} = (\hat{\psi}_{i,a})^\dagger$ only if $R$ satisfies

$$\sum_{a,b} R^{ab}_{cd}(R^{ab}_{ef})^* = \delta_{ce}\delta_{df} \quad \text{or} \quad R^{ef}_{ab} = (R^{ab}_{ef})^*. \tag{3}$$

For arbitrary $R$, a contracted bilinear operator

$$\hat{e}_{ij} = \sum_{a=1}^{m} \hat{\psi}^+_{i,a}\hat{\psi}_{j,a}, \tag{4}$$

can be constructed, which obeys

$$[\hat{e}_{ij}, \hat{\psi}^+_{k,b}] = \delta_{jk}\hat{\psi}^+_{i,b}, \tag{5a}$$

$$[\hat{e}_{ij}, \hat{\psi}_{k,b}] = -\delta_{ik}\hat{\psi}_{j,b}, \tag{5b}$$

$$[\hat{e}_{ij}, \hat{e}_{kl}] = \delta_{jk}\hat{e}_{il} - \delta_{il}\hat{e}_{kj}. \tag{5c}$$

These commutation relations will be crucial in the construction of the bosonization formulation in the subsequent section.

The single-mode partition function for an $R$-paraparticle is defined as

$$z_R(x) = \sum_{n=0}^{\infty} d_n x^n, \tag{6}$$

where $x \equiv e^{-\beta\epsilon}$, $\beta = 1/k_B T$, and $\epsilon$ is the $R$-paraparticle energy. $d_n$ are nonnegative integers and describe the dimension of the $n$-particle state. Importantly, $d_n = 0$ implies that the $n$ $R$-paraparticles cannot occupy the same mode at once. The average occupation number then can be calculated as

$$\langle n \rangle_R = -\frac{1}{\beta}\frac{\partial z_R(x)}{\partial \epsilon} = \frac{x\partial_x z_R(x)}{z_R(x)}. \tag{7}$$

For the remainder of this paper, we shall reshape the $R$-tensor as an $m$-dimensional matrix $M_{uv} = R^{ab}_{cd}$, where $u = m(a-1)+b$ and $v = m(c-1)+d$. As an example, $M$ matrices for the $m = 2$ case can be written as

$$M = \begin{pmatrix} R^{11}_{11} & R^{11}_{12} & R^{11}_{21} & R^{11}_{22} \\ R^{12}_{11} & R^{12}_{12} & R^{12}_{21} & R^{12}_{22} \\ R^{21}_{11} & R^{21}_{12} & R^{21}_{21} & R^{21}_{22} \\ R^{22}_{11} & R^{22}_{12} & R^{22}_{21} & R^{22}_{22} \end{pmatrix}. \tag{8}$$

This makes the elements of $R$ easier to visualize and allows for a convenient expression of complex $R$-tensors. In this $M$-matrix formalism, the requirement Eq. (1a) can be rewritten as

$$M^2 = 1, \tag{9a}$$

while the unitary requirement Eq. (3) becomes

$$M = M^\dagger. \tag{9b}$$

## 2.2 R-parafermions

Analogous to Green's formulation which generalizes parafermions with integer- and half-integer spins as parabosons and parafermions [5], we also construct a classification scheme for $R$-paraparticles. Suppose the GCR due to $R$ allows at most $n'$ particles to occupy a single mode, *irrespective* of particle flavor. This implies that $d_{n>n'} = 0$, hence the average occupation number is reduced to

$$\langle n \rangle_R = \frac{\sum_{n=1}^{n'} n d_n x^n}{\sum_{n=0}^{n'} d_n x^n}. \tag{10}$$

In the low temperature limit of $\beta \to \infty$,

$$x \to \begin{cases} \infty, & \epsilon < 0 \\ 0, & \epsilon > 0 \end{cases}, \tag{11a}$$

hence

$$\langle \tilde{n} \rangle_R = \begin{cases} n', & \epsilon < 0 \\ 0, & \epsilon > 0 \end{cases} = n' \theta(-\epsilon), \tag{11b}$$

i.e., the occupation number close to absolute zero is proportional to a Heaviside step function $\theta(x)$, and therefore the $R$-paraparticle possesses a fermi surface structure. This motivates us to classify $R$-paraparticles that satisfy Eq. (11b) as $R$-parafermions. Analogously, we show in the Appendix that when $d_n \neq 0$ for all $n$, the average occupation of the particle resembles bose statistics, and therefore these $R$-paraparticles are categorized as $R$-parabosons. In Green's paraparticle formulation, these behaviors were also observed in [14].

The requirement that $d_{n>n'} = 0$ implies an exclusion principle is at play within the $R$-paraparticle's GCR. In general, this can be expressed as

$$(\hat{\psi}_{i,a}^+)^p \neq 0 \quad \text{while} \quad (\hat{\psi}_{i,a}^+)^{p+1} = 0, \tag{12}$$

where $p = 1$ is the usual Pauli exclusion principle. This definition is analogous to Green's $p$-ordered parafermions, to which there can be at most $p$ particles in the same quantum state [9]. We therefore define a $p$-ordered $R$-parafermions as $R$-paraparticles that satisfy Eq. (12), i.e., allowing at most $p$ particles of the *same* flavor in a single mode. The maximum number of $R$-paraparticles that can occupy a single mode, $n'$, need not be equal to $p$.

For $m = 2$, examples of $p = 1$ $R$-paraparticles include those that are described by the $M$-matrices

$$M_1 = -\mathbb{1}_4, \tag{13a}$$

$$M_2 = \begin{pmatrix} -1 & 0 & 0 & 0 \\ 0 & 0 & \alpha & 0 \\ 0 & 1/\alpha & 0 & 0 \\ 0 & 0 & 0 & -1 \end{pmatrix}, \tag{13b}$$

and

$$M_3 = \begin{pmatrix} -1 & 1 & -1 & 0 \\ 0 & -1 & 0 & 0 \\ 0 & -2 & 1 & 0 \\ 0 & 1 & -1 & -1 \end{pmatrix}, \tag{13c}$$

where $\alpha \in \mathbb{C}_{\neq 0}$. The $R$-tensor of $M_1$ is exactly example 3 of [23] where $n' = 1$, while both $M_2$ and $M_3$ have $n' = 2$. Note that the matrices $M_2$ with $|\alpha| \neq 1$ and $M_3$ describe non-unitary

$R$-tensors, and ordinary fermions are retrieved from $\alpha = -1$ in Eq. (13b). Likewise, $p = 2$ cases also exist for $R$-paraparticles with $m = 2$, such as those described by

$$M_4 = \begin{pmatrix} 0 & 0 & 0 & \beta \\ 0 & -1 & 0 & 0 \\ 0 & 0 & -1 & 0 \\ 1/\beta & 0 & 0 & 0 \end{pmatrix}, \tag{13d}$$

also with $\beta \in \mathbb{C}_{\neq 0}$. This also describes a non-unitary $R$-tensor whenever $|\beta| \neq 1$. The exclusion statistics of $R$-paraparticles described by $M_4$ go as $\hat{\psi}_{i,a}^+ \hat{\psi}_{i,b}^+ = -\hat{\psi}_{i,a}^+ \hat{\psi}_{i,b}^+ = 0$ and $\hat{\psi}_{i,a}^+ \hat{\psi}_{i,a}^+ \propto \hat{\psi}_{i,b}^+ \hat{\psi}_{i,b}^+ \neq 0$ for $a, b \in \{1, 2\}$, $a \neq b$. This results to $\hat{\psi}_{i,a}^+ \hat{\psi}_{i,a}^+ \hat{\psi}_{i,a}^+ \propto \hat{\psi}_{i,a}^+ \hat{\psi}_{i,b}^+ \hat{\psi}_{i,b}^+ = 0$, implying that three particles cannot occupy the same mode at once, and therefore $d_{n>2} = 0$.

## 3 R-parafermion bosonization

Having defined $p$-ordering and the classification of $R$-paraparticles into $R$-parafermions or $R$-parabosons, let us now consider the bosonization of one-dimensional interacting $R$-paraparticles. We define the $R$-paraparticle Luttinger model kinetic Hamiltonian term as

$$\hat{H}_0 = \sum_{r,k,a} v_F(rk - k_F) : \hat{\psi}_{r,k,a}^+ \hat{\psi}_{r,k,a} : \tag{14a}$$

$$= \sum_{r,k} v_F(rk - k_F)\left(\hat{e}_{r,k,k} - m\theta(rk - k_F)\right) \tag{14b}$$

where the constant $\theta(rk - k_F)$ imposes the normal ordering denoted by $: \ldots :$, and $\pm k_F$ are the parafermi points of the 1D system. The indices $k$ and $a$ denote momentum and flavor, while $r \in \{+, -\}$ (a mode index) differentiates between the two branches of the dispersion relation $\varepsilon_r(k) = v_F(rk - k_F)$, with $v_F$ as the fermi velocity. In the Luttinger model, the total density operator can be decomposed in terms of the left-moving and right-moving $R$-parafermions, $\hat{\rho}(k) = \hat{\rho}_+(k) + \hat{\rho}_-(k)$, where

$$\hat{\rho}_r(q) = \sum_{k,a}\left(\hat{\psi}_{r,k+q,a}^+ \hat{\psi}_{r,k,a} - \delta_{q,0}\theta(rk - k_F)\right) \tag{15a}$$

$$= \sum_k \hat{e}_{r,k+q,q} - m\delta_{q,0}\sum_k \theta(rk - k_F) \tag{15b}$$

are the density operators for each branch. In a similar manner, flavor wave operators are defined as

$$\hat{\sigma}(q) = \hat{\sigma}_+(q) + \hat{\sigma}_-(q), \tag{15c}$$

$$\hat{\sigma}_r(q) = \sum_{k,a} \alpha_a\left(\hat{\psi}_{r,k+q,a}^+ \hat{\psi}_{r,k,a} - \delta_{q,0}\theta(rk - k_F)\right). \tag{15d}$$

For a direct comparison with the spin-$\frac{1}{2}$ Luttinger model, we restrict our results to the case of $m = 2$ and define $\alpha_1 = -1$ and $\alpha_2 = +1$ when dealing with flavor waves. Note also that Eq. (15d) cannot be expressed in terms of contracted bilinear operators defined in Eq. (4) due to the prefactor $\alpha_a$.

### 3.1 Density wave bosonization and flavor-charge separation

Using Eq. (5), the commutators of the density operator can be evaluated as

$$[\hat{\rho}_r(-q), \hat{\rho}_{r'}(q')] = \delta_{rr'}\delta_{qq'}\sum_k (\hat{e}_{r,k-q,k-q} - \hat{e}_{r,k,k}), \tag{16}$$

to which by invoking the properties of the parafermi points [28] and Eq. (11b), simplifies to

$$[\hat{\rho}_r(-q), \hat{\rho}_{r'}(q')] = r\delta_{rr'}\delta_{qq'}\frac{n'qL}{2\pi}, \tag{17}$$

($L$ being the system length) dependent only on the maximum number of particles that can occupy a single mode. Comparison with the bose commutation relation $[\hat{b}_k, \hat{b}_k^+] = \delta_{kk'}$ motivates the mapping

$$\hat{b}_q = \sqrt{\frac{2\pi}{n'qL}}\hat{\rho}_+(-q), \qquad \hat{b}_q^+ = \sqrt{\frac{2\pi}{n'qL}}\hat{\rho}_+(q),$$
$$\hat{b}_{-q} = \sqrt{\frac{2\pi}{n'qL}}\hat{\rho}_-(q), \qquad \hat{b}_{-q}^+ = \sqrt{\frac{2\pi}{n'qL}}\hat{\rho}_-(-q), \tag{18}$$

which then imply that the density waves of all $R$-parafermionic Luttinger models have bosonic properties.

Using the same commutation relations of the contracted bilinear operator, it can also be shown that

$$[\hat{\rho}_r(-q), \hat{\sigma}_{r'}(q')] = 0. \tag{19}$$

This means that for all $m = 2$ cases, density waves and flavor waves are decoupled and propagate independently, thus implying that flavor-charge separation occurs for all $R$-parafermions.

## 3.2 Flavor wave bosonization

While Eq. (19) might suggest that both $\hat{\rho}_r(q)$ and $\hat{\sigma}_r(q)$ are in general bosonic operators of different species, explicit calculation of the commutation relations of the flavor wave operators show that it is not always the case. Because we cannot express $\hat{\sigma}_r(q)$ in terms of $\hat{e}_{ij}$, we are required to evaluate

$$
\begin{aligned}
[\hat{\sigma}_r(-q), \hat{\sigma}_{r'}(q')] &= \sum_{k,k',a,a'} \alpha_a\alpha_{a'}[\hat{\psi}_{r,k-q,a}^+\hat{\psi}_{r,k,a}, \hat{\psi}_{r',k'+q',a'}^+\hat{\psi}_{r',k',a'}] \\
&= \sum_{k,k',a,a'} \alpha_a\alpha_{a'}\Big(\hat{\psi}_{r,k-q,a}^+\hat{\psi}_{r,k,a}\hat{\psi}_{r',k'+q',a'}^+\hat{\psi}_{r',k',a'} \\
&\quad - \hat{\psi}_{r',k'+q',a'}^+\hat{\psi}_{r',k',a'}\hat{\psi}_{r,k-q,a}^+\hat{\psi}_{r,k,a}\Big) \\
&= \sum_{k,k',a,a'} \alpha_a\alpha_{a'}\Big\{\delta_{r,r'}\delta_{k-q,k'}\delta_{a,a'}\hat{\psi}_{r,k-q,a}^+\hat{\psi}_{r',k',a'} \\
&\quad + \Big(\sum_{b,c,d,e,f,g} R_{a'c}^{ab}R_{ab}^{de}R_{gf}^{a'c}\hat{\psi}_{r',k'+q',d}^+\hat{\psi}_{r,k-q,e}^+\hat{\psi}_{r',k',f}\hat{\psi}_{r,k,g}\Big) \\
&\quad - \hat{\psi}_{r',k'+q',a'}^+\hat{\psi}_{r',k',a'}\hat{\psi}_{r,k-q,a}^+\hat{\psi}_{r,k,a}\Big\} \tag{20}
\end{aligned}
$$

explicitly for specific $R$-tensors. In this case, bosonic flavor waves occur only when the RHS of Eq. (20) reduces to the RHS of Eq. (16). This requirement is equivalent to

$$
\begin{aligned}
\sum_{b,c,d,e,f,g} R_{a'c}^{ab}R_{ab}^{de}R_{gf}^{a'c}\hat{\psi}_{r',k'+q',d}^+\hat{\psi}_{r,k-q,e}^+\hat{\psi}_{r',k',f}\hat{\psi}_{r,k,g} &= -\delta_{rr'}\delta_{aa'}\delta_{k-q,k'}\hat{\psi}_{r',k'+q',a'}^+\hat{\psi}_{r,k,a} \\
&\quad + \hat{\psi}_{r',k'+q',a'}^+\hat{\psi}_{r',k',a'}\hat{\psi}_{r,k-q,a}^+\hat{\psi}_{r,k,a}, \tag{21}
\end{aligned}
$$

as the second term in Eq. (21) cancels out the last term in Eq. (20) and the remaining Kronecker delta terms reduce to $\hat{e}_{ij}$. When the above requirement is satisfied, Eq. (20) becomes

$$[\hat{\sigma}_r(-q), \hat{\sigma}_{r'}(q')] = r\delta_{rr'}\delta_{qq'}\frac{n'qL}{2\pi}, \tag{22}$$

and the bosonic transformation

$$\hat{c}_q = \sqrt{\frac{2\pi}{n'qL}}\hat{\sigma}_+(-q), \qquad \hat{c}_q^+ = \sqrt{\frac{2\pi}{n'qL}}\hat{\sigma}_+(q),$$

$$\hat{c}_{-q} = \sqrt{\frac{2\pi}{n'qL}}\hat{\sigma}_-(q), \qquad \hat{c}_{-q}^+ = \sqrt{\frac{2\pi}{n'qL}}\hat{\sigma}_-(-q), \tag{23}$$

allows for the establishment of $[\hat{c}_k, \hat{c}_k^+] = \delta_{kk'}$. $R$-tensors in the form of Eq. (13a), (13b), and (13d) with $\beta^2 = 1$ all follow these results. However, $R$-parafermions described by Eqs. (13c) and (13d) with $\beta^2 \neq 1$ do not satisfy Eq. (21) and therefore their $R$-parafermionic Luttinger models do not have bosonic flavor wave profiles.

## 3.3 Equivalence spectrum

We now check wether the system described using $R$-parafermion operators are the same as that described by bosonic ones. For $m = 2$ $R$-parafermions, it can be shown that the free Luttinger model Hamiltonian obeys the commutation relation

$$[\hat{\phi}_r(q), \hat{H}_0] = v_F r q \hat{\phi}_r(q) \tag{24}$$

for $\hat{\phi}_r(q) \in \{\hat{\rho}_r(q), \hat{\sigma}_r(q)\}$. This allows us to rewrite the kinetic energy Hamiltonian in terms of number operators [26, 27],

$$\hat{H}_0 = v_F\Big[\sum_k |k|\hat{n}_k + \frac{\pi}{2L}\sum_{r,a}(\hat{N}_{r,a})^2\Big] \tag{25}$$

where $\hat{n}_k = \hat{b}_k^+ \hat{b}_k$ and $\hat{N}_{r,a} = \sum_k(\hat{\psi}_{r,k,a}^+\hat{\psi}_{r,k,a} - \theta(rk - k_F))$. Note that these results are applicable only to $R$-parafermions with bosonic flavor waves. The second summation term takes care of the normal ordering imposed in the $R$-parafermion operator basis. The flavor wave contribution to the Hamiltonian is the same as Eq. (25), but with $\hat{b}_k^+ \hat{b}_k$ replaced by $\hat{c}_k^+ \hat{c}_k$ [29], hence we consider the total Hamiltonian

$$\hat{H}_T = \hat{H}_0 + \hat{H}_{FW}, \tag{26}$$

where $H_0$ is denoted by Eqs. (14) and (25) in the $R$-parafermion and boson operator bases, respectively.

One way to verify the correspondence between these Hamiltonian forms is to show that their degeneracies match in both bases. The case where $R$-parafermions have no internal degrees of freedom (i.e., spinless fermions) has already been done in [26] by evaluating the partition functions of each basis separately. Following this method, the $R$-parafermion partition function is calculated as

$$z_{\mathrm{PF}}(y) = \prod_{n=1}^{\infty}\Big[\Big(\sum_{j=0}^{p}\frac{1}{y^{2(2n-1)(j-1)}}\Big)\Big(\sum_{j=0}^{p}y^{2(2n-1)j}\Big)\Big]^4, \tag{27}$$

where $y \equiv e^{-\beta v_F \pi/L}$. The $(2n - 1)$ exponent is obtained by assuming that the system is under periodic boundary conditions such that $k = 2n\pi/L$ (where $n$ are energy level indices) and $k_F = n_F\pi/L$. The variable $p$ denotes the order of the $R$-parafermion as discussed in Sec. 2.2. In particular, for $p = 1$, we have

$$z_{\mathrm{PF}}(y) = \prod_{n=1}^{\infty}\Big[1 + y^{2(2n-1)}\Big]^8, \tag{28a}$$

while for $p = 2$,

$$z_{\text{PF}}(y) = \prod_{n=1}^{\infty}\left[\left(y^{2(2n-1)} + 1 + y^{-2(2n-1)}\right)\left(1 + y^{2(2n-1)} + y^{4(2n-1)}\right)\right]^4. \qquad (28b)$$

Meanwhile, the boson partition function is

$$Z_B(y) = \left[\sum_{n=-\infty}^{+\infty}(y)^{n^2}\right]^4\prod_{n=1}^{\infty}\left(\frac{1}{1-y^{4n}}\right)^4 = \prod_{n=1}^{\infty}\left[\frac{(1+y^{(2n-1)})^2(1-y^{2n})}{(1-y^{4n})}\right]^4, \qquad (29)$$

the second equality being a result of the elliptic theta function equivalence

$$\theta_3(0, w) = \sum_{m=-\infty}^{+\infty}w^{m^2} = \prod_{n=1}^{\infty}(1+w^{2n-1})^2(1-w^{2n}). \qquad (30)$$

Comparison between Eqs. (28a) and (29) shows that in the low-temperature limit (i.e., $\beta \to \infty$, and subsequently $y \to 0$), the boson and $p = 1$ $R$-parafermion partition functions coincide, hence the spectrum of the free Luttinger model is equivalent in the $R$-paraparticle and boson operator basis only if the $R$-parafermion obeys Pauli's exclusion principle. This result generalizes the idea that the Luttinger model is acceptable only at low-energies [46]. On the other hand, if the $R$-parafermion possesses a higher-order exclusion principle (such as in the case of $p = 2$ in Eq. (28b)), $z_{PF}$ quickly diverges. This means that bosonization is not applicable for $R$-parafermions with $p > 1$, and possibly the Luttinger model is not appropriate for modelling such systems.

## 4 Potential experimental realization

The Luttinger model assumes a linearized single-particle dispersion which makes it valid only for particles with momentum close to $\pm k_F$. In the fermionic Luttinger model literature, all ideal one-dimensional systems exhibit spin-charge separation, but for realistic materials these conditions are not always met and systems that satisfy the above requirements are said to reside in the Luttinger liquid phase. Several works have experimentally verified the existence of the Luttinger liquid phase by observing some of its key signatures [31,47–51]. We highlight a potential experimental design for observing $R$-paraparticle signatures by borrowing ideas from [31], which we outline as follows.

In the above paper, Luttinger liquid signatures were detected as two parabolic dispersions due to separate fermi seas with different masses, which are associated with spin and charge excitations. Since $R$-parafermionic Luttinger models also exhibit flavor-charge separation, we should expect that a one-dimensional system containing $R$-parafermions would also display such behavior. Current predictions show that $R$-paraparticles can occur as quasiparticle excitations [23]. In this case, the fermi momentum in one dimension, expressed as [55]

$$k_F = \frac{N_0 \pi}{n'L}, \qquad (31)$$

is in general not equal for different $R$-parafermions. Here, $N_0$ denotes the number of particles occupying the ground state, and assuming that the system is made up of fundamental fermions which in turn have emergent $R$-parafermion quasiparticle properties, the number of fermions in the ground state is generally not equal to the number of $R$-parafermions in the quasiparticle ground state. By then constructing a system which does not exhibit spin-density excitations (e.g., there is no spin-charge separation or the system is fully spin-polarized) but contains

emergent $R$-paraparticles that are in the Luttinger liquid phase, then signatures of flavor-charge separation may also be detected as separate dispersion relations as in [31].

The above proposal is roughly sketched, but it does give us a roadmap of research extensions to undertake in order to achieve an experimental observation of $R$-paraparticle signatures through Luttinger liquids. Firstly, a method which can turn a system of ordinary fermions or bosons to quasiparticle $R$-parafermion excitation is required. In [23], it was shown that emergent one-dimensional quasiparticle excitations with parastatistical GCRs can be developed from a generic spin model where the spin operators have $R$-paraparticle behaviors. For example, emergent $R$-paraparticles described by Eq. (13a) were shown to be realizable using three-level Rydberg atom or molecule systems [56, 57], while those described by Eq. (13b) with $\alpha = 1$ can be modelled using systems described in [58]. In general, several papers have proposed a method of mapping between particles of different statistics in one-dimensional systems [59, 60], although an extension to the $R$-paraparticles is yet to be explored.

Nonlinear extensions of the fermionic Luttinger model [46, 61, 62] have also shown that spin-charge separation can persist outside the previously predicted low-energy regime [31]. While this may allow for easier detection of flavor wave oscillations at higher temperatures, it may also induce spin-charge separation at the same time. We are then required to find parameter regimes where spin-wave excitations cease to exist while flavor waves persist.

Further complications to this proposal would be due to the limitations of our results to $R$-paraparticles with 2 internal degrees of freedom. The construction of the flavor wave operator was inspired using an analog of spin waves in spin-$\frac{1}{2}$ systems. While quantum spins are not directly observable, electron spin-wave propagation can be observed experimentally [42]. As such, Eq. (15d) should be sufficient to describe a wave propagation due to some internal degree of freedom of an $R$-paraparticle with $m = 2$. This allows us to potentially observe $R$-paraparticle signatures even without full knowledge of the internal symmetries at play within the system. However, if the emergent $R$-paraparticle contains $m > 2$ internal degrees of freedom, then the flavor wave operator must be redefined to account for multiple types of flavor wave oscillations, while at the same time, preserving the total degrees of freedom of the system.

## 5   Summary and Discussion

We have shown that bosonization applies to certain one-dimensional $R$-paraparticles systems described by the Luttinger model, to which flavor-charge separation occurs as a result. These $R$-paraparticles are classified as $R$-parafermions, which obey general exclusion principles and thus can only occupy a limited number of particle states. $R$-parafermionic density waves are shown to have bosonic signatures, but flavor waves generally do not exhibit this behavior. A comparison of the partition function in the Luttinger model shows that only $R$-parafermions that satisfy Pauli's exclusion principle preserve their degeneracy when changing from the $R$-parafermion basis to the boson basis, further limiting the applicability of bosonization to $R$-paraparticles of order $p = 1$. Nonetheless, these results show promises in potentially observing $R$-paraparticle signatures in one-dimensional systems.

We have motivated this study as a means of finding a general means of solving an $R$-paraparticle system in the presence of interaction. A dedicated section for such method was omitted in this paper since, by showing that a collection of $R$-parafermion operators can be recast as a boson operator, bosonization automatically implies that solutions applicable to fermionic Luttinger models should also apply to $R$-parafermionic systems (see Sec. 4.4C of [29]). Nonetheless, other generalizations such as the field-theoretic formalism of the Luttinger model should not be treated as trivial given the extensive use of generalized commutation

relations in their definitions.

As mentioned in the previous section, generalization to arbitrary $m$ cases would benefit the development of possible experimental setups that may result to observations of $R$-paraparticle systems. This goes hand-in-hand with an understanding of the internal symmetries at play within the $R$-paraparticle. At the same time, a physical interpretation of a non-bosonic flavor wave oscillation is lacking, which may or may not affect potential observations of $R$-paraparticle singatures. Speculatively, this could also mean that $\hat{\sigma}_r(q)$ generally possess parabose statistics, but this remains an assumption until an example with such properties can be presented.

## Acknowledgements

We thank M.A. Sulañgi for the helpful discussions.

**Funding information**   D.F.S. acknowledges support from the DOST-SEI under the Accelerated Science and Technology Human Resource Development Program.

## A   R-parabosons

In Sec. 2.2, we discussed the behavior of $R$-paraparticles where $d_{n>n'} = 0$ for some finite $n'$. We now discuss what happens in the limit $n' \to \infty$. We note that the single-mode partition functions of $R$-paraparticles satisfy [23]

$$z_R(-x)z_{-R}(x) = 1 \tag{A.1}$$

under the premise that $z_R(x)$ is interpreted as the Hilbert series [63] of $R$. Suppose $R$ describes an $R$-parafermion, then by Eq. (A.1),

$$z_{-R}(x) = \frac{1}{\sum_{n=0}^{n'} d_n(-x)^n} = \sum_{j=0}^{\infty} D_j x^j, \tag{A.2}$$

where the second equality follows the definiton of Eq. (6). The coefficients $D_j$ can be obtained by a geometric series expansion of $1/z_R(-x)$ and are expressed as

$$D_0 = 1, \quad D_j = \sum_{k=1}^{\min(j,n')} d_k(-)^{k+1} D_{j-k} \in \mathbb{Z}_+. \tag{A.3}$$

Eq. (A.2) applies only in the reduced domain $x \in (0, e^{-\beta \epsilon_0})$ under the following considerations:

- By definition, $z_R(-x) = 1 + \sum_{n=1}^{n'} d_n(-x)^n$ ($d_0 = 0$ for all $R$-paraparticles, i.e., there is always only one vacuum state), so the second equality in Eq. (A.2) requires $1 > |\sum_{n=1}^{n'} d_n(-x)^n|$. $1 > -\sum_{n=1}^{n'} d_n(-x)^n$ is automatically satisfied by the fact that $z_{-R}(x) > 0$ as required from a partition function.

- The minimum particle energy must be finite, i.e., $\epsilon_0 \not\to -\infty$, as in this limit, $x \to \infty$, and

$$\lim_{x \to \infty} \sum_{n=1}^{n'} d_n(-x)^n = \begin{cases} +\infty, & \text{for even } n' \\ -\infty, & \text{for odd } n' \end{cases}. \tag{A.4}$$

The geometric expansion requires $\sum_{n=1}^{n'} d_n(-x)^n < 1$, which is violated by the above equation, thus, $\epsilon_0$ must be finite.

- Because $z_{-R}(e^{-\beta\epsilon_0}) \not> 0$, then $z_{-R}(x)$ must diverge at $x = e^{-\beta\epsilon_0}$. Note that $z_{-R}(e^{-\beta\epsilon_0}) \neq 0$ because $z_R(-x)$ is a smooth function and therefore does not diverge for some finite $x$. This implies that $e^{-\beta\epsilon_0}$ is one of the roots of $z_R(-x)$.

From these, the average occupation number of $-R$ is therefore

$$\langle\tilde{n}\rangle_{-R} = \frac{\sum_{n=1}^{n'} n d_n (-)^{n-1} x^n}{\sum_{n=0}^{n'} d_n (-x)^n} = \frac{x\partial_{-x} z_R(-x)}{z_R(-x)}. \tag{A.5}$$

By Rolle's theorem, any function will always have more real roots than their derivative, therefore, $z_R(-x)$ has at least one more zero than $\partial_{-x} z_R(-x)$, and thus $\langle\tilde{n}\rangle_{-R}$ will always be divergent in at least one $\epsilon$ value. There are no jump discontinuities in $\langle\tilde{n}\rangle_{-R}$, and removable discontinuities can always be factored out. Moreover,

$$\lim_{\beta\epsilon\to\infty} \langle\tilde{n}\rangle_{-R} = 0, \tag{A.6a}$$

$$\partial_{\beta\epsilon} \langle\tilde{n}\rangle_{-R}\Big|_{\beta\epsilon\to\infty} \propto -d_1, \tag{A.6b}$$

hence at the domain $\epsilon \in (\epsilon_0, \infty)$, the average occupation number is exponentially decreasing from infinity, reminiscent of bose statistics. This motivates us to classify $R$-paraparticles that have single-mode partition functions in the form of Eq. (A.2) with $D_n \neq 0$ for all $n$ as $R$-parabosons.

While all $R$-parafermions with a four-tensor $R$ have a corresponding $R$-parabosons described by $-R$, the converse is not always true. Consider the examples

$$M_5 = \begin{pmatrix} \alpha & 0 & 0 & 0 \\ 0 & 0 & \gamma & 0 \\ 0 & \gamma & 0 & 0 \\ \beta & 0 & 0 & -\alpha \end{pmatrix}, \tag{A.7a}$$

$$M_6 = \begin{pmatrix} \alpha & 0 & 0 & \beta \\ 0 & 0 & \gamma & 0 \\ 0 & \gamma & 0 & 0 \\ 0 & 0 & 0 & -\alpha \end{pmatrix}, \tag{A.7b}$$

and

$$M_7 = \begin{pmatrix} \alpha & 0 & 0 & 0 \\ 0 & 0 & \delta & 0 \\ 0 & 1/\delta & 0 & 0 \\ 0 & 0 & 0 & -\alpha \end{pmatrix}, \tag{A.7c}$$

where $\alpha, \gamma = \pm 1$, $\beta \in \mathbb{C}$, $\delta \in \mathbb{C}_{\neq 0}$. In all cases, no exclusion principle can be extracted from their GCRs, and therefore all possible combinations of $\alpha, \beta, \gamma$, and $\delta$ would result to $d_n \neq 0$ for all $n$. Moreover, their average occupation number

$$\langle\tilde{n}\rangle = \frac{xz'(x)}{z(x)} = \frac{\sum_{n=1}^{\infty} n d_n x^n}{\sum_{n=0}^{\infty} d_n x^n} \tag{A.8}$$

will always diverge at the radius of convergence $x_0$ of $z(x)$. This is because $xz'(x)$ grows faster than $z(x)$ as $x \to x_0^-$, and thus the resulting behavior of $\langle\tilde{n}\rangle$ is $R$-parabosonic. An implication of this result is that in the $R$-paraparticle formulation, there are more $R$-parabosons than $R$-parafermions.

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
