# Peer review of "Bosonization in $R$-paraparticle Luttinger models"

_SciPost Physics Core_

## Round 2 · Referee Report · Anonymous (Referee 1) · 2025-12-4

Report

The authors consider one-dimensional systems of paraparticles in a formulation recently introduced by Wang and Hazzard. They are interested in the question if such systems can be bosonized and what the bosonized Hamiltonian looks like. In addition to the consistency requirement from Wang, Hazzard which leads to the constant Yang-Baxter equation, the authors introduce an additional condition, Eq. (12), as in Green's construction of parafermions which is a higher order Pauli principle.

In my view, there are significant problems both with the premise of this work as well as with the results.

1) The authors never clearly distinguish between the algebra and the representation. The algebra alone only guarantees consistency but by itself does not define a physical model. For the p=2 example in Eq. (13d) the algebra only shows that there cannot be an 'a' and a 'b' particle on the same site, therefore also excluding occupancies with n>2. However, the algebra only shows that Psi^dagger_a Psi^dagger_a \propto Psi^dagger_b Psi^dagger_b and not, as written in the paper, that these states are non-zero. This is an additional assumption related to the concrete representation of the model. The algebra itself is also consistent with Psi^dagger_a Psi^dagger_a \propto Psi^dagger_b Psi^dagger_b = 0. A concrete p=2 model is never constructed in the paper.

My general point is that it is not sufficient to just consider the algebra. If one wants to discuss how to bosonize a model, one has to choose a concrete representation which includes, in particular, a local Hilbert space. I therefore believe that the premise of the entire paper is incomplete.

2) For the p=1 case, the authors obtain the partition function (28a), which is the known result for spinless fermions. This seems to imply that any p=1 parafermion model leads to this partition function. However, the flavor degrees of freedom and the concrete representation of the model will modify the partition function. With multiple flavors and a Hamiltonian which does not explicitly depend on the flavor degrees, as assumed here, the reordering of the flavors for the periodic boundary conditions considered here will lead to a separation of the partition function into flux sectors as well as to a non-zero ground state entropy (macroscopic degeneracy).

3) For the p=2 case, the authors find a partition function (28b) which diverges exponentially for T -> 0. For T->0 the partition function, appropriately normalized, should instead behave as Tr(exp(-beta(H-E_0))) -> 1. This seems to be more than just a normal ordering issue. Also, it is again unclear to which concrete p=2 model this partition function belongs to. As for the p=1 case, not all p=2 models will have the same partition function.

Overall, I do not think that one can meaningfully address bosonization and low-energy CFT structures based on the algebra of the paraparticles alone, without specifying a concrete representation. Furthermore, the flavor degrees seem to be absent from the partition functions and the partition function for p=2 is exponentially diverging for T->0 and thus does not describe a physical model. These issues undermine the main conclusions of the paper regarding bosonization, the central charge, and the concrete structure of the low-energy effective theory.

Recommendation

Reject

---

## Editorial Decision

voting_in_preparation